# A Reference-Free Lossless Compression Algorithm for DNA Sequences Using a Competitive Prediction of Two Classes of Weighted Models

**DOI:** 10.3390/e21111074

**Published:** 2019-11-02

**Authors:** Diogo Pratas, Morteza Hosseini, Jorge M. Silva, Armando J. Pinho

**Affiliations:** 1Institute of Electronics and Informatics Engineering of Aveiro, University of Aveiro, 3810-193 Aveiro, Portugal; seyedmorteza@ua.pt (M.H.); jorge.miguel.ferreira.silva@ua.pt (J.M.S.); ap@ua.pt (A.J.P.); 2Department of Electronics, Telecomunications and Informatics, University of Aveiro, 3810-193 Aveiro, Portugal; 3Department of Virology, University of Helsinki, 00100 Helsinki, Finland

**Keywords:** lossless data compression, DNA sequences, competitive prediction, weighted models, context models, stochastic repeat models

## Abstract

The development of efficient data compressors for DNA sequences is crucial not only for reducing the storage and the bandwidth for transmission, but also for analysis purposes. In particular, the development of improved compression models directly influences the outcome of anthropological and biomedical compression-based methods. In this paper, we describe a new lossless compressor with improved compression capabilities for DNA sequences representing different domains and kingdoms. The reference-free method uses a competitive prediction model to estimate, for each symbol, the best class of models to be used before applying arithmetic encoding. There are two classes of models: weighted context models (including substitutional tolerant context models) and weighted stochastic repeat models. Both classes of models use specific sub-programs to handle inverted repeats efficiently. The results show that the proposed method attains a higher compression ratio than state-of-the-art approaches, on a balanced and diverse benchmark, using a competitive level of computational resources. An efficient implementation of the method is publicly available, under the GPLv3 license.

## 1. Introduction

The arrival of high throughput DNA sequencing technology has created a deluge of biological data [1]. With the low sequencing costs of next-generation sequencing [2], metagenomics [3], ancient genomes [4], and biomedical applications [5], the number of available complete genomes is increasing widely. Most of the data are discarded and, when classified as crucial, compressed using general or specific purpose algorithms. Additionally, with the increasing of ancient sequenced genomes, the quantity of data to be compressed is now achieving a higher magnitude [6,7].

There are many file formats to represent genomic data—for example, FASTA, FASTQ, BAM/SAM, VCF/BCF, and MSA, and many data compressors to represent specifically these formats [8,9,10,11,12,13,14,15,16,17,18,19,20,21,22,23]. All of these file formats have in common the genomic sequence part, although in different phases or using different representations. The ultimate aim of genomics, before downstream analysis, is to assemble high-quality genomic sequences, allowing for having high-quality analysis and consistent scientific findings.

Genomic (or DNA) sequences are codified messages, from an alphabet of four symbols Θ={A,C,G,T}, containing instructions, structure, and historical marks of all known cellular organisms [24]. Initially, genomic sequences were compressed with general-purpose algorithms or tools, such as gzip (www.gzip.org), bzip2 (http://sourceware.org/bzip2), or LZMA (www.7-zip.org/sdk.html). Since the emergence of BioCompress [25], the development of specific compression algorithms for these sequences revolutionized the field.

The development of specific compression algorithms of DNA sequences has now 27 years. There are many lossless compression algorithms explicitly developed for genomic sequences [26]. These algorithms rely on a trade-off between compressibility and computational resources. The reasons are the specific need to balance the program with hardware characteristics and the compression purpose. Industrial-oriented compression purposes, often aiming for ultra-fast and low memory consumption computations at the expense of poor compressibility, are different from the scientific-oriented approach of developing novel models, usually aiming for better compressibility at the expense of higher computational resources.

The nature of the specific data compressors takes advantage of the inclusion of sub-programs to efficiently handle specific DNA properties, namely a high number of copies, a high level of substitutional mutations, high heterogeneity, and multiple rearrangements, such as inverted repeats [27,28]. Additionally, genomic sequences may contain data from other sources—for example, environmental factors [29,30], exogeneous inclusion [31,32], and unknown sources [33]. Compressing genomic sequences requires the ability to model heterogeneous, dynamic, incomplete, and imperfect information [34].

The lossless compression of genomic sequences has been addressed using two approaches: reference-based and reference-free [35]. The reference-based approach usually achieves substantially better compression results, mainly because when two sequences are almost identical, a model that efficiently represents the differences of one according to the other achieves top compression results. Using variations of this methodology, several reference-based approaches were proposed [8,36,37,38,39,40,41,42,43,44,45,46,47,48].

Although the reference-based approach is exceptionally efficient, there is the need to store a reference sequence. On the other hand, the reference-free approach has the advantage of not needing any reference. This approach is essential for reducing the storage of reference sequences and, more importantly, to estimate the quantity of information contained in a DNA sequence, an approximation of the Kolmogorov complexity [49]. The latter has proven to be able to infer insights into genomic evolution as well as being suitable to group sequences by nature [50].

Despite the usage of general-purpose algorithms with heavy computational models, such as neural networks, to compress genomic sequences, the specific compressors that efficiently address the specific characteristics of genomic sequences show higher compression capabilities (5–10%) using substantially less computational resources [51].

As depicted in Figure 1, the first specific algorithm is Biocompress. The Biocompress is based on a Lempel–Ziv dictionary approach [52], exploring repeats and palindromes. The Biocompress2 [53] is an extension of Biocompress [25], adding arithmetic coding of order-2 as a fallback mechanism.

The Cfact algorithm [54] uses parsing, where exact repeats are loaded in a suffix tree along with the positions indexes and encoding. The CDNA [55] was the first algorithm to combine statistical compression with approximate repeat for DNA compression. The ARM algorithm [56] explores the probability of a subsequence by summing the probabilities given the explanations of how a subsequence is generated. The ARM and CDNA algorithms yield significantly better compression ratios than older algorithms.

The offline algorithm [57] models repeated regions for compression. Only exact repeats are considered during each iteration, and the algorithm selects a substring that leads to the contraction suffix tree used to find the substring with the maximum possible number of non-overlapping occurrences. The GenCompress algorithm [58] explores the existence of approximate repeats. The DNACompress [59] finds approximate repeats using the PatternHunter [60] and, then, uses a Lempel–Ziv approach [52] for encoding. The CTW + LZ algorithm [61] is based on the context tree weighting method, which uses a weighting of multiple models to determine the next symbol probabilities. The algorithm detects approximate repeats using dynamic programming and then encodes long exact and approximate repeats using an LZ77-type encoding. Short repeats and non-repeats are encoded using a CTW.

The NMLComp algorithm [62] uses the normalized maximum likelihood (NML) model to encode approximate repeats using discrete regression and, then, combines it with a first-order context model. The GeNML [63] presents an improvement to NMLComp method, namely restricting the approximate repeats matches to reduce the cost of search, choosing the block sizes for parsing the target sequence, and uses scalable forgetting factors for the memory model.

The DNASequitur [64] is a grammar-based compression algorithm which infers a context-free grammar to represent the input sequence. Besides the exact repeats, it recognizes inverted repeats when creating rules and during substitutions. The DNA-X algorithm [65] takes advantage of repetitions by searching and encoding exact and approximate repeats. The approach uses much lower computational resources than older algorithms while achieving a competitive compression ratio. The DNAC [66] is an update of the Cfact algorithm working in four phases. It builds a suffix tree to locate exact repeats; all exact repeats are extended into approximate repeats by dynamic programming; it extracts the optimal non-overlapping repeats from the overlapping ones; it uses the Fibonacci encoding method to encode the repeats in a self-delimited way. The DNAPack algorithm [67] works by finding approximate repeats to encode them optimally. It uses a dynamic programming approach for the selection of the segments.

The XM algorithm [68] is still one of the most successful compressors given its compression capabilities at the expense of more computational resources, both time and memory. It combines three types of models: repeat models, a low-order context model, and a short memory context model of 512 bytes. The probabilities are encoded using arithmetic coding.

The Differential Direct Coding algorithm (2D) [69] uses side information strategies to accommodate large data sets using multiple sequences and auxiliary data. 2D is suitable for any sequence data, including substantial length data sets, such as genomes and meta-genomes. The DNASC algorithm [70] compresses the DNA sequence horizontally, first by using extended Lempel–Ziv style, and, then, vertically by taking a block size equal to 6, and a window size equal to 128. The GBC algorithm [71] assigns binary bits in a preprocessing stage to exact and reverse repeat fragments of DNA sequences. The DNACompact algorithm [72] uses a preprocessing edition of the bases for after representation using encoding by word-based tagged code (WBTC) [73]. The POMA tool [74] uses particle swarm optimization, which makes the algorithm feasible only for tiny sequences.

The DNAEnc3 algorithm [75] uses a competition of context models of several depths (orders up to sixteen) and, then, redirects the probabilities to an arithmetic encoder. It uses sub-programs embedded in the context models to handle the inverted repeats. The DNAEnc4v2 algorithm [76], instead of competition as in DNAEnc3, the context models are combined with a soft-blending mechanism that uses a particular decaying forgetting factor to give importance to the context models that achieve better performance.

The LUT algorithm [77] uses a four-step coding rule. It includes the use of a LUT (Look Up Table), character transformations, tandem repeats handling, and segment decisions. The GenCodex algorithm [78] yields a better compression ratio at high throughput by using graphical processing units (GPUs) and multi-core in two phases, namely bit-preprocessing and fragment representation using either one or two bytes.

The BIND algorithm [79] adopts a unique ’block-length’ encoding for representing binary data. BIND also handles other symbols than ACGT. The DNA-COMPACT algorithm [80] exploits complementary contextual models to search for exact repeats and palindromes and represent them by a compact quadruplet. Then, it uses non-sequential contextual models where the predictions of these models are synthesized using a logistic regression model. The HighFCM algorithm [81] explores a pre-analysis of the data before compression to identify regions of low complexity. This strategy enables the use of deeper context models (context order up to 32), supported by hash-tables, without requiring huge amounts of memory. The SeqCompress algorithm [82] uses a statistical model and, then, arithmetic coding to compress DNA sequences.

Transforming genomic sequences into images, where a two-dimensional space substitutes the one-dimensional space, is an approach that is used in [83,84]. In [83], firstly, the Hilbert space-filling curve is exploited to map the target sequence into an image. Secondly, a context weighting model is used for encoding the image. In [84], the CoGI (Compressing Genomes as an Image) algorithm is presented, which initially transforms the genomic sequence into a binary image (or bitmap), then, uses a rectangular partition coding method [85] to compress the image and, finally, explores entropy coding for further compression of the encoded image and side information.

The GeCo algorithm [86] uses a soft-blending cooperation with a specific forgetting factor between context models and substitutional tolerant context models [87] before employing arithmetic encoding. It has sub-programs to deal with inverted repeats and uses cache-hashes for deeper context models. In GeCo2 [88], the mixture of models is enhanced, where each context model or tolerant context model now has a specific decay factor. Additionally, specific cache-hash sizes and the ability to run only a context model with inverted repeats are available.

The OCW algorithm [89] uses an optimized context weighting based on the minimum description length and the least-square algorithm for the optimization of the weights. The OBComp algorithm [90] uses a single bit to code the two highest occurrence nucleotides. The positions of the two others are saved. Then, it uses a modified version of an RLE technique and the Huffman coding algorithm.

In this paper, we propose a new algorithm (Jarvis) that uses a competitive prediction based on two different classes: Weighted context models and Weighted stochastic repeat models. The Weighted context models use a soft-blending mechanism, with a decaying forgetting factor, of context and substitutional tolerant context models. The Weighted stochastic repeat models also uses a soft-blending mechanism, with a decaying forgetting factor, between multiple repeat models of specific word size. Both classes use sub-programs to handle inverted repeats. The competitive prediction is based on the highest probability of each class at a precise moment. The model is trained along with the prediction using a context model. The final probabilities, for each base, are coded using an arithmetic encoder.

This paper is organized as follows. In the next section, we describe the compressor in detail. Then, we present the comparative results of the proposed compressor against state-of-the-art algorithms in a fair and consistent benchmark proposed in [91]. The latter includes a discussion of some possible development lines. Finally, we make some conclusions.

## 2. Method

The method is based on a competitive prediction between two classes of models: Weighted context models and Weighted stochastic repeat models. As depicted in Figure 2, for each prediction, the probabilities are redirected to an arithmetic encoder. The context models (at the left of Figure 2) are combined through a weighted set of context and substitutional tolerant context models [86,87] using a specific forgetting factor for each model, while the Weighted stochastic repeat models (at the right of Figure 2) use a common forgetting factor.

The method enables setting any number of context models and repeat models, as long as at least one model is used. This setup permits very high flexibility to address different types of DNA sequences and creates space for further optimization algorithms.

In the following subsections, we describe in detail the Weighted context models, the Weighted stochastic repeat models, the competitive prediction model, and the implementation of the algorithm. For the purpose, we assume that there is a source generating symbols from a finite alphabet Θ, where Θ={A,C,G,T}. We also assume that the source has already generated the sequence of *n* symbols $x^{n}=x_{1}x_{2}\cdots x_{n},\;x_{i}\in \Theta$. Therefore, a subsequence of $x^{n}$, from position *i* to *j*, is denoted as $x_{i}^{j}$.

### 2.1. Weighted Context Models

Context models are finite statistical models assuming the Markov property. A context model of an information source assigns probability estimates to the symbols of the alphabet, according to a conditioning context computed over a finite and fixed number, *k*, of past outcomes (order-*k* context-model) [92]. A substitutional tolerant context model (STCM) [86,87] is a probabilistic-algorithmic context model. It acts as a short program that enables setting the number of allowed substitutions in a certain context depth. In practice, it assigns probabilities according to a conditioning context that considers the last symbol, from the sequence to occur, as the most probable, given the occurrences stored in the memory instead of the true occurring symbol. An STCM, besides being probabilistic, is also algorithmic, namely because they can be switched on or off given its performance, according to a threshold, *t*, defined before the computation [86]. The threshold enables or disables the model, according to the number of times that the context has been seen, given *l* hits or fails that are constantly stored in memory in a cache array. For both context models and STCMs, the number of conditioning states of the model is ${\left|\Theta \right|}^{k}$ (in our case, $4^{k}$).

We assume that the memory model starts with counters all set to zero. Through all the computation, the memory model is updated according to the outcomes. Therefore, the prediction of each context model is set along with the training. Notice that, in Figure 2, the models four and five share the same memory model (CM4) because model five is an STCM with the same *k* as in model four.

The cooperation of both context models and STCM is supervised by a soft blending mechanism [75,93] that gives importance to the models that had a better performance, given an exponential decaying memory [75]. Figure 2 depicts an example of the cooperation between four context models and one substitutional tolerant context model, $C_{i},i=1,\ldots ,5$. Each of these models, $C_{i}$, has a probability (CP), a weight (CW), and a memory model (CM) associated with it.

For a model, the probability of each symbol, $x_{i}$, is given by
(1)$$P\left(x_{i}\right)=\sum_{m\in M}P_{m}\left(x_{i}|x_{i-k}^{i-1}\right)\;w_{m,i},$$ where $P_{m}\left(x_{i}|x_{i-k}^{i-1}\right)$ is the probability assigned to the next symbol by a context model or STCM, *k* is the order of the corresponding model *m*, and where $w_{m,i}$ denotes the corresponding weighting factor, with
(2)$$w_{m,i}\propto {\left(w_{m,i-1}\right)}^{\gamma_{m}}P_{m}\left(x_{i}|x_{i-k}^{i-1}\right),$$ where the sum of the weights, for each respective model, is constrained to one, and where $\gamma_{m}\in \left[0,1\right)$ acts as a forgetting factor for each model. We found experimentally that models with lower *k* are related to lower $\gamma_{m}$ (typically, below 0.9), while higher *k*, is associated with higher $\gamma_{m}$ (near 0.95). This means that, in this mixture type, the forgetting intensity should be lower for more complex models. A curious indication was also found for a context model of order six. This model seems to be efficient with $\gamma_{m}\in \left[0.75,0.85\right]$ and is associated with k=6, which is the lowest $\gamma_{m}$ among the models. We hypothesize that this might be related to the period multiplicity found in the DNA [94].

The depth of the model, *k*, identifies the number of contiguous symbols seen in the past for predicting the next symbol and, hence, $x_{i-k}^{i-1}$ [92]. We use an estimator parameter (alpha) that allows for balancing between the uniform and the frequency distribution (usually the deepest models have lower alphas [81]).

Inverted repeats are essential to consider because they can give additional compression gain [95]. Therefore, we use a short program that allows mapping for subsequences with similarity to inverted repeat sequences according to the algorithm of [96].

The cache-hash [97] enables keeping in mind only the latest entries up to a certain number of hash collisions. This is very important because the memory models of the deepest context models have very sparse representations and, hence, storing its entries in a table would require $4^{k+1}$ entries, which means that assuming counters of 8 bits for a k=20 would require 4 TB of RAM. A linear hash would be feasible, depending on the available RAM and sequence size. In order to remove space constraints, we set a maximum number of collisions, enabling us to maintain a maximum peak of RAM.

### 2.2. Weighted Stochastic Repeat Models

The repeat model, also known as a copy expert from the XM compression method [68], is a model that stores into memory the positions relative to the sequence that has an identical *k*-mer identified in the past of the sequence. The positions are stored, using a causal processing paradigm, usually in a memory model as a hash-table. The model is used after a *k*-mer match occurs and is switched off after a certain threshold of performance is reached.

Figure 3 depicts an example of a repeat model with *k*-mer size of 8 while Figure 2 (at the right side) represents the architecture. The positions of where the subsequence occurred in the past are stored in the hash table. In this example, two positions are identified, namely 14,251 and 14,275. If we used only one position, this would be similar to the GReEn implementation [41]. However, we use the information at most from RPN models (RPN are the maximum number of repeats models which are shown in Figure 2). When the RPN is higher than the available number of positions, the number of actual models is bounded by the maximum.

These repeat models are called stochastic because, to start a new repeat (after a *k*-mer match), any position given the same *k*-mer (in the hash table) has the same probability of being used. If we used the sequential order, the initial positions of the sequence would be more used, given the number of repeats being upper bounded by RPN. Therefore, the stochastic nature enables uniform distribution of the repeats to start in different positions along the sequence. Another advantage is the absence of indexes to represent the position of the repeat being used under the positions vector. As such, the stochastic nature allows decreasing the memory inherent to the representation of the hash table.

Along with the hash table of positions, the sequence needs to be continuously preserved in memory (both in compression and decompression). To minimize its representation in memory, we pack each DNA symbol into two bits, instead of the common 8 bits. This approach allows for decreasing to a factor of four the memory associated with the representation of the sequence. Notice that sequences with length 100 MBases would require 100 MBytes of RAM just to be represented. With the packing approach, only 24 MB are needed.

The repeat models are combined using the same methodology in the context models. For each repeat, there is a weight which is adapted according to its performance. In this case, the decaying ($\gamma_{m}$) is very small since the weights need to be quickly adapted.

### 2.3. Competitive Prediction Context Model

The competitive prediction is used to choose the best-predicted class of models between Weighted context models and Weighted stochastic repeat models. The prediction is modeled using a context model with a specific order-size defined as a parameter. The context model uses a binary alphabet, where each symbol corresponds to a different class. The sequence of symbols containing the best model is represented by *Z*, where $Z_{i}$ is a symbol of the sequence at a given instant *i*.

Figure 4 depicts an example of a competitive prediction context model with a context order depth of five. To predict the best class of models to represent $Z_{i+1}$, the probability of $P(Z_{i+1}=S|k)$ needs to be computed, having *S* as the next symbol (in this case S=0) and *k* as the context order depth with the previous five symbols.

The class that has the highest probability of being used is the one which will be used to model a specific base. Accordingly, the probabilities of the selected class will be forwarded to the arithmetic encoder. After that, the information of the best class is updated into the context model of the classes.

Since in the CPCM the context order (*k*) is the crucial parameter, we assessed the impact of the variation of the context order according to different modes for different genomic sequences. Figure 5 depicts this assessment using the HoSa, EnIn, AeCa, and YeMi sequences (in decreasing order of sequence length). The remaining plots for the other sequences in the dataset can be found at the code repository. Generally, there is a relation between the context order of the CPCM and the length of the sequence (according to the respective redundancy), where longer sequences require a higher context order, and shorter sequences stand for lower context orders. As an example, the HoSa sequence (largest) is better compressed (in level 12) with a context order of 16, while the YeMi sequence (shortest) is better compressed (in level 2) with a context order of 5.

The described competitive prediction model runs in high-speed using reasonable accuracy. The accuracy of the model can improve with the development of a prediction based on multiple models, namely through Weighted context models. However, this creates a trade-off between accuracy and computational time, which may be very high for the gains that it may produce.

### 2.4. Decompression

For a compression method to be considered lossless, all the compressed sequences must be decompressed exactly to the original sequences. The current compression and decompression methodologies are symmetric. This symmetry means that both Weighted context models, Weighted stochastic repeat models, and competitive prediction model are synchronized in the same order using the same characteristics. Additional side information is included in the compressed file (in the beginning) in order for the decompressor to use the same characteristics. For example, in the Weighted stochastic repeat models, the seed is passed in the header to ensure the exact beginning in the stochastic process.

Accordingly, all the files used in this article have been losslessly decompressed using the same machine and OS (Linux Ubuntu). Regarding different floating-point hardware implementations, we have only tested one sequence compression–decompression (DrMe) between different hardware and OS version, namely compressing with one (server) machine and with a specific OS and, then, decompress with a different (Desktop) machine and OS version. Although it has worked in this example, we can not guarantee that it stays synchronized on machines if they have different floating-point hardware implementations.

### 2.5. Implementation

The tool (Jarvis), written in C language, is available at https://github.com/cobilab/jarvis, under the GPL-v3 license, and can be applied to compress/decompress any genomic sequence. The alphabet of the sequences is truncated to ACGT symbols. We use a slightly modified implementation of an Arithmetic Encoder provided by Moffat et al. [98].

The tool is accompanied with the appropriate decompressor, which uses slightly less time to decompress than to compress, and approximately the same RAM. The decompressor is approximately symmetric. All the sequences that we tested have been losslessly decompressed.

The tool includes several default running modes from 1 to 15. Apart from some exceptions, lower levels use less computational resources (time and RAM) and are more prone to shorter sequences, while higher levels work better in larger sequences. Nevertheless, specific model configurations can be manually set as parameters to the program.

## 3. Results

In this section, we benchmark the proposed compressor against state-of-the-art data compressors. The dataset proposed for this benchmark contains 15 genomic sequences [91], with a consistent balance between the number of sequences and sizes. Moreover, it reflects the main domains and kingdoms of biological organisms, enabling a comprehensive and balanced comparison. The dataset contains a total DNA sequence lenght of 534,263,017 bases (approximately half a GigaByte).

Ranking the algorithms mentioned in the Introduction is a complex task. For example, some of these algorithms have been a contribution to other extensions or applications, while others are specialized for specific types of genomic sequences, such as bacteria, collections of genomes, and alignment data. There are also algorithms to cope with low computational resources. From our experience, we would highlight XM [68] and GeCo/GeCo2 [86,88] given their ability to compress genomic sequences with high compression ratios. On average, XM is slightly better concerning compression ratio (approximately 0.4% and 0.2% over GeCo and GeCo2, respectively). However, XM uses substantially higher RAM and time than GeCo and GeCo2 [88,91]. An algorithm that uses substantially lower computational resources is CoGI [84]; however, it is less efficient in the compaction. From a large number of available specific genomic data compressors, we choose XM, GeCo, GeCo2, and CoGI for a benchmark with Jarvis. In addition, we include two general-purpose compressors, namely LZMA and PAQ8 (both using the best compression parameters).

The results presented in this paper can be reproduced, under a Linux OS, using the scripts provided at the repository https://github.com/cobilab/jarvis, specifically scripts/Run.sh. The results of GeCo2 have been imported from [88].

We present the comparative results of the proposed compressor (Jarvis) against state-of-the-art algorithms. Table 1 depicts the number of bytes needed to compress each DNA sequence for each compressor and Table 2 the computational time. As can be seen, on average, Jarvis compressed the dataset to the lowest number of bytes. In some sequences, GeCo2 and XM were able to achieve better compression, although with a minimal difference. Jarvis uses pre-set levels to compute and, hence, these values may be improved with higher levels and optimization.

Regarding PAQ8 (in its best compression option), Jarvis achieves a compression improvement of 5.2%, approximately. This comparison is according to the measures in [51]. In addition, on average, Jarvis is faster 140 times than PAQ8.

Regarding CoGI, the method provides a small factor of compressibility, better than gzip 2.3% (although not present in the table, gzip in the best option achieve 150,794,589 bytes). Nevertheless, CoGI is the fastest method. On average, CoGi is faster than Jarvis 28 times, although Jarvis achieved 31% higher compression ratio. CoGI is more suitable for industry-orientation purposes.

Jarvis shows and improvement of 1.1% and 0.9% to GeCo and GeCo2, respectively. The computational time is competitive with GeCo and slighty more than GeCo2. Regarding the second-best tool in compression ratio (XM), Jarvis improves the compression to approximately 0.6%. In addition, it is faster 5.7 times more than XM. Regarding RAM, Jarvis used a maximum peak of 7.12 GB in the largest sequence. These are competitive memory values with GeCo/GeCo2 and, at least, half of the RAM needed by XM.

Figure 6 shows the compressed size and speed, where the mean of speed values for all datasets is calculated to obtain the average speed for each method. As depicted, Jarvis shows the best compression rate since the compressed size is the lowest. On the other hand, GeCo, GeCo2, and XM seem to have very similar performance, while PAQ8 and LZMA are not so efficient in genomic data. Regarding the speed, Jarvis is approximately at the level of LZMA and GeCo2, showing that the trade-off between computational resources and precision is minimal.

Additionally, Jarvis can run with other modes. In Figure 7, we include a comparison of all the fifteen modes in Jarvis for the three largest sequences. For example, running Jarvis with level 12 in HoSa sequence achieves **38,280,246** bytes (1.6139 BPS). This result is an improvement of 1% over Jarvis in mode 7. The trade-off is computational time and RAM, however still less than XM. Therefore, Jarvis is flexible and can be optimized to achieve considerably better compression ratios. The optimization, besides the choice of the best model, can be applied in a specific combination of the number of models, depths, estimator parameters, among many others.

One of the main achievements of this paper is to combine Weighted context models with Weighted stochastic repeat models using a competitive prediction model. In order to test the impact of the inclusion of both repeat models and competitive prediction model, we include a very repetitive sequence (exogenous from the benchmarking dataset). This test has the underlying idea that repetitive regions are better modeled with Weighted stochastic repeat models than by Weighted context models. The test sequence is the assembled human Y-chromosome downloaded from the NCBI.

As depicted in Figure 8, all the modes from Jarvis compress better than the best mode from GeCo2. Jarvis (mode 12) achieves a compression 5.413% better than GeCo2 (mode 15) using approximately the same computational time. This example shows a substantial improvement using both Weighted stochastic repeat models and the competitive prediction model.

## 4. Conclusions

The development of efficient DNA sequence compressors is fundamental for reducing the storage allocated to projects. The importance is also reflected for analysis purposes, given the search for optimized and new tools for anthropological and biomedical applications.

In this paper, we presented a reference-free lossless data compressor with improved compression capabilities for DNA sequences. The method uses a competitive prediction model to estimate, for each symbol, the best class of models to be used before applying arithmetic encoding. The method uses two classes of models: Weighted context models (including substitutional tolerant context models) and Weighted stochastic repeat models. Both classes of models use specific sub-programs to handle inverted repeat subsequences efficiently.

The results show that the proposed method attains a higher compression ratio than state-of-the-art approaches using a fair and diverse benchmark. The computational resources needed by the proposed approach are competitive. The decompression process uses approximately the same computational resources.

## Figures and Tables

**Figure 1 entropy-21-01074-f001:**
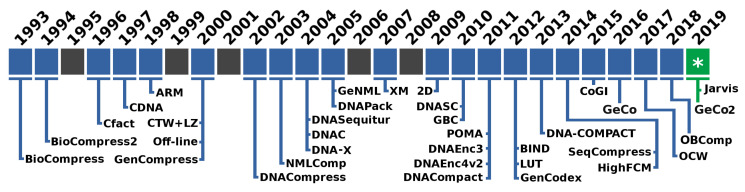
Timeline with the names of the proposed data compressors specifically for genomic sequences.

**Figure 2 entropy-21-01074-f002:**
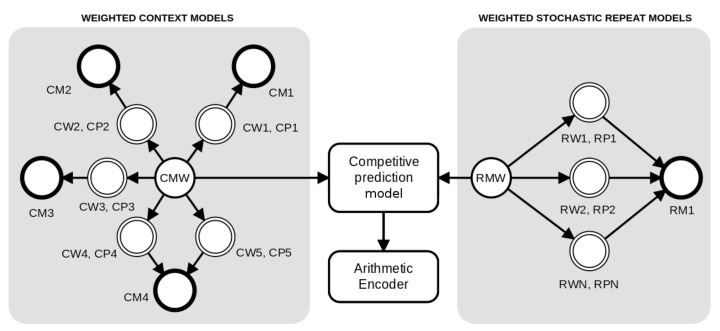
An architecture example of a competitive prediction between five Weighted context models (at left, represented with prefix *C*) and three Weighted stochastic repeat models (at right, represented with prefix *R*). Each model has a weight (*W*) and associated probabilities (*P*) that are calculated according to the respective memory model (*M*), where the suffix complements the notation. The tolerant context model (CW5,CP5) uses the same memory of model four (CW4,CP4), since they have the same context. Independently, the probabilities of the context models and repeat models are averaged according to the respective weight and redirected to the competitive prediction model. Finally, the probabilities of the model class with the highest probability (predicted) are redirected to the arithmetic encoder.

**Figure 3 entropy-21-01074-f003:**
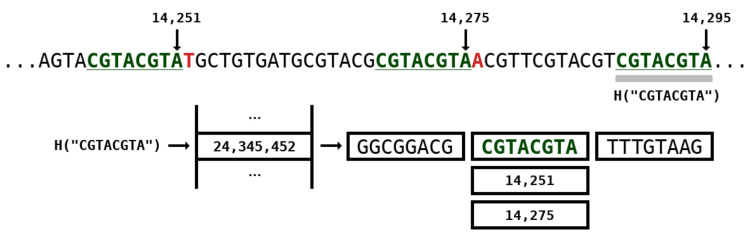
Repeat model example with *k*-mer size of 8. The *H* is a hash function that encapsulates a *k*-mer into a natural number on the hash table. Positions 14,251 and 14,275 stand for identical *k*-mers seen in the past of the sequence. Number 14,295 stands for the current position of the base being coded.

**Figure 4 entropy-21-01074-f004:**
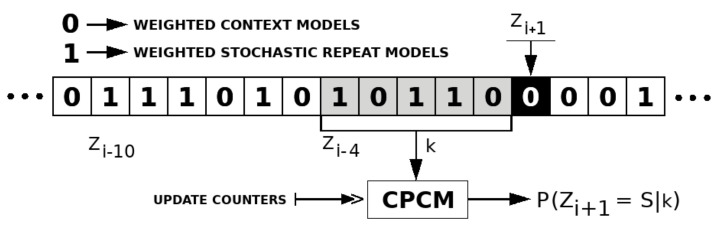
Competitive prediction context model (CPCM) example with context depth (*k*) of 5. The next symbol is *S*, and *Z* is the sequence with the best class of models estimated by the CPCM.

**Figure 5 entropy-21-01074-f005:**
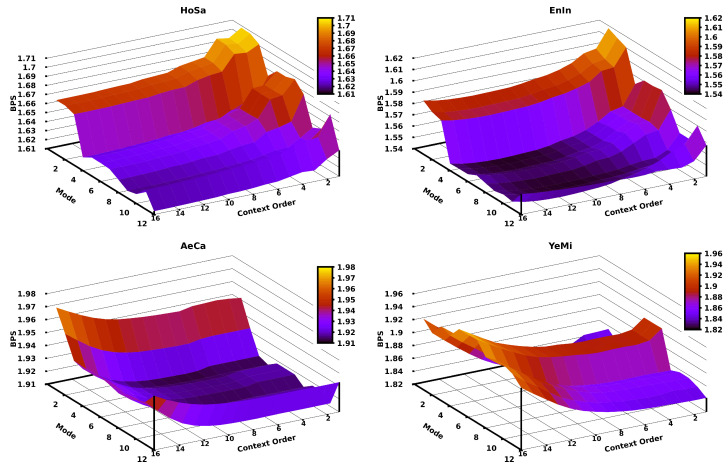
Bits per base (BPS) of compressing four sequences applying a CPCM context order variation for the first twelve modes of Jarvis. The four datasets are sorted according to different sizes; namely, the largest is HoSa (**left**-**top**), then, EnIn (**right**-**top**), AeCa (**left**-**bottom**), and YeMi (**right**-**bottom**).

**Figure 6 entropy-21-01074-f006:**
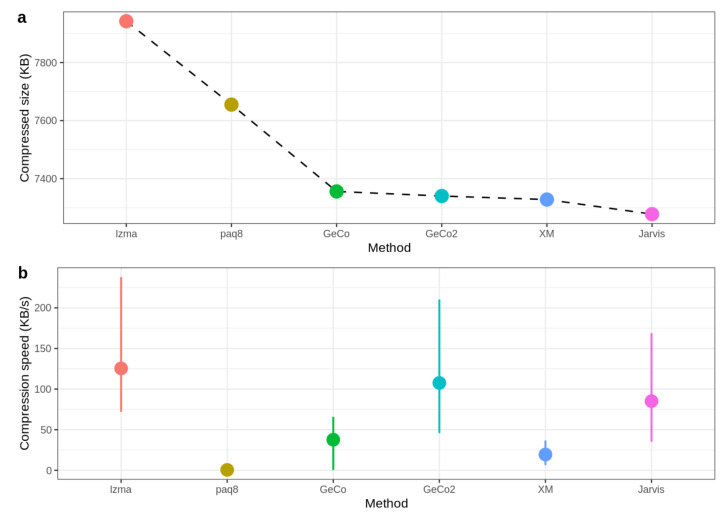
Benchmark with size (**a**) and speed (**b**). For each sequence, the value of speed is calculated as compressed size (KB) divided by compression time (s). The mean of speed values for all datasets is calculated to obtain the average speed for each method. The CoGI compressor is not included because it is an outlier concerning this dataset.

**Figure 7 entropy-21-01074-f007:**
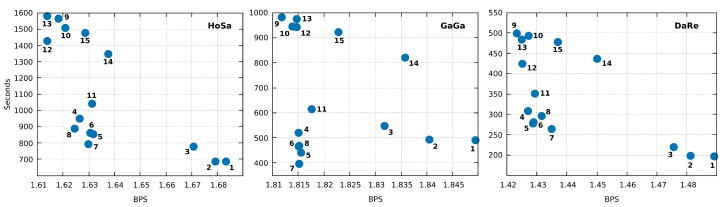
Comparison of the fifteen compression modes available in Jarvis for the three largest sequences in the dataset (HoSa, GaGa, and DaRe). Compression ratios are in Bits Per Symbol (BPS) and Time in seconds. Times may not agree precisely with Table 2 because we rerun the tool. Each number, corresponding to the blue dots, stands for the mode/level used in Jarvis. We recall that additional levels or specific configurations can be set.

**Figure 8 entropy-21-01074-f008:**
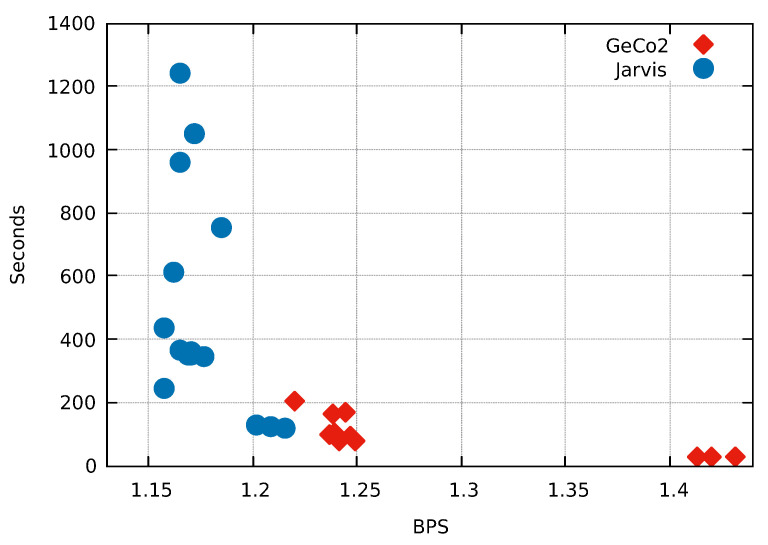
Comparison of the fifteen compression modes available in Jarvis and GeCo2 for the human chromosome Y sequence. Compression ratios are in Bits Per Symbol (BPS) and Time in seconds. Each number, corresponding to the blue dots, stands for the mode/level used in the respective compressor.

**Table 1 entropy-21-01074-t001:** Number of bytes needed to represent each DNA sequence given the respective data compressor (LZMA -9, PAQ8 -8, CoGi, GeCo, XM and Jarvis). We ran LZMA with the -9 flag (best option), PAQ8 with the -8 (best option), GeCo using “-tm 1:1:0:0/0 -tm 3:1:0:0/0 -tm 6:1:0:0/0 -tm 9:10:0:0/0 -tm 11:10:0:0/0 -tm 13:50:1:0/0 -tm 18:100:1:3/10 -c 30 -g 0.9”, GeCo2 with parameters from [88], and XM using 50 copy experts. The compression level used in Jarvis is depicted between parentheses, and it has been set according to the size of the sequence. The length of the sequences is present in Table 2.

ID	LZMA-9	PAQ8-8	CoGI	GeCo	GeCo2	XM	Jarvis (level)
HoSa	42,292,440	40,517,624	51,967,817	38,877,294	38,845,642	38,940,458	**38,660,851** (7)
GaGa	36,179,650	34,490,967	40,846,177	33,925,250	33,877,671	33,879,211	**33,699,821** (6)
DaRe	12,515,717	12,628,104	17,084,450	11,520,064	11,488,819	11,302,620	**11,173,905** (5)
OrSa	9,348,183	9,280,037	11,999,580	8,671,732	8,646,543	8,470,212	**8,448,959** (5)
DrMe	8,016,544	7,577,068	8,939,690	7,498,808	**7,481,093**	7,538,662	7,490,418 (5)
EnIn	5,785,343	5,761,090	7,210,867	5,196,083	5,170,889	5,150,309	**5,087,286** (4)
ScPo	2,722,233	2,557,988	2,921,247	2,536,457	2,518,963	2,524,147	**2,517,535** (4)
PlFa	2,097,979	1,959,623	2,411,342	1,944,036	1,925,726	1,925,841	**1,924,430** (4)
EsCo	1,185,704	1,107,929	1,307,943	1,109,823	1,098,552	1,110,092	**1,095,606** (4)
HaHi	985,096	904,074	1,124,483	906,991	902,831	913,346	**899,464** (3)
AeCa	413,886	380,273	454,357	385,640	**380,115**	387,030	380,507 (3)
HePy	415,161	385,096	457,859	381,545	375,481	384,071	**374,362** (3)
YeMi	19,262	16,835	19,805	17,167	**16,798**	16,861	16,861 (2)
AgPh	12,183	10,754	12,243	10,882	**10,708**	10,711	10,745 (2)
BuEb	5441	4668	5291	4774	4686	**4642**	4690 (1)
Total	121,994,822	117,582,130	146,763,151	112,986,546	112,744,517	112,558,213	**111,785,440**

**Table 2 entropy-21-01074-t002:** Computational time (in seconds) needed to represent each DNA sequence given the respective data compressor (LZMA, PAQ8, CoGi, GeCo, GeCo2, XM, and Jarvis). We ran LZMA with the -9 flag (best option), PAQ8 with the -8 (best option), GeCo using “-tm 1:1:0:0/0 -tm 3:1:0:0/0 -tm 6:1:0:0/0 -tm 9:10:0:0/0 -tm 11:10:0:0/0 -tm 13:50:1:0/0 -tm 18:100:1:3/10 -c 30 -g 0.9”, GeCo2 with parameters from [88], and XM using 50 copy experts. The compression level used in Jarvis is depicted between parentheses and it has been set according to the size of the sequence. The length scale of the sequences is in bases.

ID	Length	LZMA	PAQ8	CoGI	GeCo	GeCo2	XM	Jarvis
HoSa	189,752,667	552.5	85,269.1	25.2	648.6	652.4	5,589.8	814.8 (7)
GaGa	148,532,294	468.7	64,898.9	19.9	503.2	494.7	3,633.9	412.3 (6)
DaRe	62,565,020	170.0	29,907.7	8.2	215.9	198.8	785.2	284.9 (5)
OrSa	43,262,523	112.9	20,745.1	5.8	192.4	138.3	489.7	234.5 (5)
DrMe	32,181,429	85.6	14,665.8	4.3	114.6	102.4	362.6	66.7 (5)
EnIn	26,403,087	66.0	11,183.6	3.7	95.8	82.5	279.8	101.1 (4)
ScPo	10,652,155	23.0	4,619.1	1.5	45.2	34.2	96.5	28.7 (4)
PlFa	8,986,712	18.3	4,133.9	1.2	39.7	35.3	84.4	25.4 (4)
EsCo	4,641,652	8.1	1,973.9	0.6	26.4	5.1	36.8	10.9 (4)
HaHi	3,890,005	6.9	1,738.1	0.5	23.7	4.4	39.1	7.1 (3)
AeCa	1,591,049	2.2	675.3	0.2	17.0	1.9	10.3	2.2 (3)
HePy	1,667,825	2.3	715.1	0.2	17.2	1.9	11.2	2.7 (3)
YeMi	73,689	0.1	32.6	0.0	12.3	0.1	0.9	0.2 (2)
AgPh	43,970	0.0	20.1	0.0	12.1	0.1	0.9	0.1 (2)
BuEb	18,940	0.0	9.1	0.0	12.2	0.1	0.7	0.1 (1)
Total	534,263,017	1516.6	240,587.4	71.3	1976.3	1742.2	11,421.8	1991.7

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
