# Peer review of "A Reference-Free Lossless Compression Algorithm for DNA Sequences Using a Competitive Prediction of Two Classes of Weighted Models"

_entropy, 2019, doi:10.3390/e21111074_

Round 1
Reviewer 1 Report
Necessary content edits: r. 108: space missing: "POMA tool[73]"r. 197: "We found, by exhaustive search"
The nature of the search is not clear, and should be indicated.
If it was experimental, adding "We found experimentally," would suffice.
r. 297 onwards: The names paq8 and lzma are written in lowercase throughout the paper. Both should be written in uppercase.
r. 297/8: "Jarvis achieves the lowest number of bytes to compress the dataset."
Bytes are not a resource used to compress data, therefore the sentence should be rephrased, e.g.:
"Jarvis compressed the dataset to the lowest number of bytes"
r. 302/3: "Jarvis is faster 140 times than paq8."
Adding "on average" is strongly suggested.
r. 306/7: "CoGi is faster than Jarvis 28 times, although Jarvis compresses 40% more"
Adding "on average" is strongly suggested.
Moreover, Jarvis did not "compress 40% more" as the volume of input data was the same, it rather "achieved 31% higher compression ratio". Suggested extension:
"Additionally, Jarvis can run with optimized modes."
I believe including a scatter graph depicting results (at least in terms of compression time and achieved compressed size, in respective axes) of different Jarvis modes (and, for comparison, the reference methods) on a single file (e.g. of medium size) would be valuable to comprehend the processing time/compression effectiveness trade-off of the respective modes.
Author Response
Reviewer 1:
Thank you for taking the time to review this manuscript.
Necessary content edits:
r. 108: space missing: "POMA tool[73]"
Done. Thanks for pointing out.
r. 197: "We found, by exhaustive search"
The nature of the search is not clear, and should be indicated.
If it was experimental, adding "We found experimentally," would suffice.
Done. Thanks for pointing out.
r. 297 onwards: The names paq8 and lzma are written in lowercase throughout the paper. Both
should be written in uppercase.
Done. Thanks for pointing out.
r. 297/8: "Jarvis achieves the lowest number of bytes to compress the dataset."
Bytes are not a resource used to compress data, therefore the sentence should be rephrased, e.g.:
"Jarvis compressed the dataset to the lowest number of bytes"
Done. Thanks for pointing out.
r. 302/3: "Jarvis is faster 140 times than paq8."
Adding "on average" is strongly suggested.
Done. Thanks for pointing out.
r. 306/7: "CoGi is faster than Jarvis 28 times, although Jarvis compresses 40% more"
Adding "on average" is strongly suggested.
Moreover, Jarvis did not "compress 40% more" as the volume of input data was the same, it rather
"achieved 31% higher compression ratio".
The reviewer is correct. It is now edited (31%). Thanks for pointing out.
Suggested extension:"Additionally, Jarvis can run with optimized modes."
I believe including a scatter graph depicting results (at least in terms of compression time and
achieved compressed size, in respective axes) of different Jarvis modes (and, for comparison, the
reference methods) on a single file (e.g. of medium size) would be valuable to comprehend the
processing time/compression effectiveness trade-off of the respective modes.
Thank you for the idea. We now include an additional image for comparison of Jarvis using other
modes for the three more extensive sequences (HoSa, GaGa, DrMe). We have included the largest
ones because these are the ones that have a higher impact on the overall results.

Reviewer 2 Report
The manuscript describes a new technique for the compression of DNA sequences. The authors propose the use of multiple context models, which are then employed based on their previous performance. --------------- Regarding the writing ---------------------- First of all, the article requires some serious proofreading. A non-exhaustive enumeration of writing issues: - 'If particular'; abstract line 2. - Review hyphenation on compound adjectives: "competitive-prediction", "substitutional tolerant", ... - It is 'low costs' that are associated with 'new sequencing technologies', and not the other way around. - Do not start a line with a citation. - 'Quantify of data'. - 'higher volume of magnitude'. - l38 (i.e., line 38) 'is' should be 'are' and the rest of the sentence should agree in number. - 'high copy number'. - 'other alteration sources' bzip2? Which algorithm is 7zip? Provide citations for these algorithms (https://www.ietf.org/rfc/rfc1951.txt ?). - Did really BioCompress 'revolutionized the field'? - Same notation is employed for x^n and w_{m,i-1}^... - l178 CM5 is not present in figure 2. CM4 -> CM4/CM5. - 'The t enables'. 't' -> 'threshold'. - After Eq. 1, CM includes STCM. - l196 for all -> for each. - 'prone to use' is not the right phrasing. - l199 why should it be lower? - In the equation in l201, do not use ';', nor '->'. - 'The H' 'The class that has the highest probability estimate of being used is the one which will be used'. - 'the same RAM', 'higher RAM'. - Table 2, units? Sections 2.1 and 2.2 could be written in a more understandable manner. For example, section 2.1, starts describing how cooperation is supervised without having introduced how the cooperation concept is employed. Then it proceeds to describe the context models while at times assuming the reader is unfamiliar with the concept and, at times, assuming the reader is familiar with the subject at hand (pick one and be consistent). --------------- Regarding the method ---------------------- There are couple of issues that may need to be addressed regarding the actual contents of the manuscript. - If the entropy encoder that is being employed is the one here https://people.eng.unimelb.edu.au/ammoffat/arith_coder/, it may contain additional context modeling which may interfere with the results. - How is it guaranteed that context models stay synchronized on machines with different floating point hardware implementations? - There is no analysis of the predictor of context model. Which are the error rates of the predictor of context model? Which context models are used the most? Are they used in runs? Why is the predictor not used to select among context models? --------------- Overall impression ---------------------- The benefits of Jarvis over GeCo2 are extremely small, more so if taking into account that Jarvis seems to be an improvement to GeCo2. Without sufficient analysis on the predictor and given the small performance improvement, I would recommend not to publish the manuscript in its current state.Author Response
Reviewer 2:
The manuscript describes a new technique for the compression of DNA sequences. The authors
propose the use of multiple context models, which are then employed based on their previous
performance. -
Thank you for taking the time to review this manuscript.
One of the main achievements of this paper is to combine context models with Weighted Stochastic
Repeat Models (WSRM) using a competitve prediction model. Both the competitive prediction
model and WSRM are new and, as far as we know, never have been described. We also provide an
efficient implementation (in C and without any dependencies, besides the arithmetic encoder),
which enables us to test any configuration of the models, although we have included 15 modes (we
added three more) of pre-computed models. Using only seven modes, we consistently show better
average results using competitive computational resources. Nevertheless, the tool/method is very
flexible in terms of parameterization, which allows choosing a different number of models, both
context models, and stochastic repeat models. Therefore, it allows future works on optimization.
-------------- Regarding the writing ----------------------
First of all, the article requires some serious proofreading. A non-exhaustive enumeration of writing
issues:
- 'If particular'; abstract line 2.
Done. Thanks for pointing out.
- Review hyphenation on compound adjectives: "competitive-prediction", "substitutional tolerant",
...
Done. Thanks for pointing out.
- It is 'low costs' that are associated with 'new sequencing technologies', and not the other way
around.
We changed the sentence. Thanks for pointing out.
- Do not start a line with a citation.
We tried to comply as much as possible with this style of writing. Thanks for pointing out.
- 'Quantify of data'.
Done. Thanks for pointing out.- 'higher volume of magnitude'.
Done. Thanks for pointing out.
- l38 (i.e., line 38) 'is' should be 'are' and the rest of the sentence should agree in number.
Done. Thanks for pointing out.
- 'high copy number'.
Done. Thanks for pointing out.
- 'other alteration sources'
Done. Thanks for pointing out.
bzip2? Which algorithm is 7zip?
Corrected. Thanks for pointing out.
Provide citations for these algorithms (https://www.ietf.org/rfc/rfc1951.txt ?).
We now include the URL of each algorithm/tool and obviously removed 7z (given the LZMA
presence). Thanks for detecting it.
- Did really BioCompress 'revolutionized the field'?
We consistently believe in this.
Biocompress was the first data compressor dedicated to biological sequences, which makes a big
difference in the way it is approached. Regardless of the compression gains that the tool has, it has
inspired many authors (including us) to develop specific models to address this type of data. One of
the results of the article shows a big difference between general-purpose data compressors (such as
PAQ8) and Jarvis. BioCompress started this revolution.
- Same notation is employed for x^n and w_{m,i-1}^...
They mean different things: W_{m,i-1} is the previous weight associated with a specific model,
while x^n has to do with a sequence with a specific size.
- l178 CM5 is not present in figure 2. CM4 -> CM4/CM5.
We corrected the notation to C, where CM is the memory model (M) of a context model (C).
Thanks for pointing out.
- 'The t enables'. 't' -> 'threshold'.
Done. Thanks for pointing out.
- After Eq. 1, CM includes STCM.
Corrected. Thanks for pointing out.
- l196 for all -> for each.
Done. Thanks for pointing out.
- 'prone to use' is not the right phrasing.
Done. Thanks for pointing out.
- l199 why should it be lower?
The lower k context models adapt faster and, hence, they must have a higher forgetting intensity
associated.
- In the equation in l201, do not use ';', nor '→'.
Done. Thanks for pointing out.
- 'The H' 'The class that has the highest probability estimate of being used is the one which will be
used'.
We did not understand this remark.
- 'the same RAM', 'higher RAM'.
“approximately the same RAM” is what it represents.
- Table 2, units?
Done. Thanks for pointing out.
Sections 2.1 and 2.2 could be written in a more understandable manner. For example, section 2.1,
starts describing how cooperation is supervised without having introduced how the cooperation
concept is employed. Then it proceeds to describe the context models while at times assuming the
reader is unfamiliar with the concept and, at times, assuming the reader is familiar with the subject
at hand (pick one and be consistent).
Done. Thanks for pointing out.
--------------- Regarding the method ----------------------
There are couple of issues that may need to be addressed regarding the actual contents of the
manuscript. - If the entropy encoder that is being employed is the one here
https://people.eng.unimelb.edu.au/ammoffat/arith_coder/, it may contain additional context
modeling which may interfere with the results.
- How is it guaranteed that context models stay synchronized on machines with different floating
point hardware implementations?
- There is no analysis of the predictor of context model. Which are the error rates of the predictor of
context model? Which context models are used the most? Are they used in runs? Why is the
predictor not used to select among context models?
We are aware of the possible different floating-point implementation issues.
The arithmetic encoder is not set to use context modeling since we provide this before and, then,
renormalize to integer all the estimations. Therefore, this may only happen at the modeling phase
(given floating-point operations).
We have only tested one sequence compression-decompression (DrMe) between different hardware
and OS version, namely compressing with one (server) machine and with a specific OS and, then, decompress with a different (Desktop) machine and OS version. It has losslessly decompressed inthis example. However, we can not guarantee that it stays synchronized on machines if they have
different floating-point hardware implementations.
We have also included this information in the manuscript.

Reviewer 3 Report
Authors proposed an idea about the reference-free lossless compression algorithm.
Recently, we have tons of large-scale sequence data that should be compressed, but it is not always good to compress data. Because it needs time to un-compress when we need it.
Basically, I agree with the proposed idea, which is simple but good enough to show its performance.
The overall write-up seems to be good enough in a way to address to solve the problem.
Thanks.
Author Response
Thank you for taking the time to review this manuscript.

Reviewer 4 Report
Authors developed a new lossless compressor "jarvis" for DNA sequences with competitive-prediction model. Using two models, they achieved a higher compression ratio than exist approaches.
Majore:
It seems that a very convenient compression algoritm has been developed. However, additional information is requred regarding the presentation and discussion of the benchmark resutls. For example, OrSa (43 Mbp) has been compressed to 8. 44 Mb (compression rate = 19%), while DrMe (32 Mbp) has been done to 7.49 M (23%). Is this due to differences in sequence properties? If the difference in the ratio depends on the sequence property, it is better to divide the genome sequence into several segments instead of full-length, compress each segment, and compare the average compression ratio.
Minor:
It would be nice to have the abbrivation list about ID (DrMe, OrSa, etc).
Author Response
Reviewer 4:
Authors developed a new lossless compressor "jarvis" for DNA sequences with competitive-
prediction model. Using two models, they achieved a higher compression ratio than exist
approaches.
Thank you for taking the time to review this manuscript.
Majore:
It seems that a very convenient compression algoritm has been developed. However, additional
information is requred regarding the presentation and discussion of the benchmark resutls. For
example, OrSa (43 Mbp) has been compressed to 8. 44 Mb (compression rate = 19%), while DrMe
(32 Mbp) has been done to 7.49 M (23%). Is this due to differences in sequence properties? If the
difference in the ratio depends on the sequence property, it is better to divide the genome sequenceinto several segments instead of full-length, compress each segment, and compare the average
compression ratio.
Thanks for pointing out. The OrSa corresponds to Oriza Sativa, which is a plant while DrMe to
Drosophila (fly). Yes, but not only the distribution of the GC content. Jarvis can address small and
significant alterations in the data efficiently. Dividing the genome into segments would remove the
ability to model repetitive patterns and, hence, would decrease the capability to compress de
sequence efficiently.
Minor:
It would be nice to have the abbrivation list about ID (DrMe, OrSa, etc).
Done. Thanks for pointing out.

Round 2
Reviewer 1 Report
Thanks for the correction. I did not spot any new issues.
Reviewer 4 Report
My comments are basically reflected, so i think this paper is suitable for publish.